# Isolation and Purification, Structural Characterization and Antioxidant Activities of a Novel Hetero-Polysaccharide from Steam Exploded Wheat Germ

**DOI:** 10.3390/foods11091245

**Published:** 2022-04-26

**Authors:** Lei Hu, Xiaodan Zhou, Xue Tian, Ranran Li, Wenjie Sui, Rui Liu, Tao Wu, Min Zhang

**Affiliations:** 1State Key Laboratory of Food Nutrition and Safety, Tianjin Key Laboratory of Food Quality and Health, College of Food Science and Engineering, Tianjin University of Science & Technology, Tianjin 300457, China; huleistar@126.com (L.H.); dawnzhou5@163.com (X.Z.); xuetian9701@163.com (X.T.); li17862686224@126.com (R.L.); lr@tust.edu.cn (R.L.); wutaoxx@gmail.com (T.W.); 2China-Russia Agricultural Processing Joint Laboratory, Tianjin Agricultural University, Tianjin 300392, China

**Keywords:** wheat germ polysaccharide, steam explosion, isolation and purification, modification, structural characterization, antioxidant activities

## Abstract

A purified polysaccharide, designated as SE-WGP_I_, was isolated from wheat germ modified by steam explosion. The primary structure characteristics were determined by HPGPC, GC, periodate oxidation-Smith degradation, methylation analysis, FT-IR, NMR and Congo red test. The results showed that SE-WGP_I_ was a homogeneous hetero-polysaccharide with the average molecular weight of 5.6 × 10^3^ Da. The monosaccharide composition mainly consisted of glucose, arabinose and xylose with a molar ratio of 59.51: 20.71: 19.77. The main backbone of SE-WGP_I_ consisted of →4,6)-α-D-Glc*p*(1→6)-α-D-Glc*p*(1→3)-β-D-Xyl*p*(1→5)-α-L-Ara*f*(1→ and the side chain was α-D-Glc*p*(1→ linked at the C4-position of →4,6)-α-D-Glc*p*(1→. SE-WGP_I_ likely has a complex netted structure with triple helix conformation and good thermal stability. In addition, SE-WGP_I_ had valid in vitro radical scavenging activities on DPPH and hydroxyl radicals. This study may provide structural information of SE-WGP_I_ for its promising application in the fields of functional foods or medicines.

## 1. Introduction

Wheat germ is a highly nutritive milling by-product of wheat processing. About 2.5 million tons of it are produced per year around the world [1]. Wheat germ is considered to be the most nutritious part of wheat grain, providing 409 kcal per 100 g, with 54% provided by carbohydrates, 20% by lipids and 26% by proteins. Many studies have demonstrated that wheat germ has an important effect on human health (such as antihyperlipidemic, anticancer, antioxidant and hypocholesterolemic effects) [2]. Meanwhile, studies on wheat germ are mainly focused on the thermal and non-thermal modification of wheat germ, functional properties of polysaccharide and protein hydrolysates, encapsulation of wheat germ oil, and relevant applications in flour products. Although an abundant amount of wheat germ is removed during processing and widely used in animal feed, its commercial use still remains minimal and results in resource waste [3].

Since carbohydrates are the major component (56.3 g/100 g) in wheat germ, a significant amount of polysaccharides (18.4 g/100 g) has been found [4]. Recently, plant-derived polysaccharides have drawn wide attention owing to their abundant pharmacological activities and biodegradable, naturally non-toxic and biocompatible properties. In our previous study [5], the primary structure of polysaccharides isolated from wheat germ were identified, involving molecular weight, monosaccharide composition, functional groups, glycosidic linkage and branching characteristics. However, the structural and functional activities of polysaccharides could be strongly influenced by various modification treatments of materials. Recent studies have attempted to prepare polysaccharides using extrusion cooking, high-pressure homogenization, enzymatic hydrolysis and fermentation, etc. Several structural features modified by the above methods, including monosaccharide composition, functional groups, molecular weight, conformation and branching, etc., have been identified as responsible for various activity promotions of specific polysaccharides. Furthermore, not only the individual, but also combinations of these structural characteristics seem to influence the direct correlation between polysaccharides and cells or other components of the immune system.

Among many industrial technologies, steam explosion (SE) is a typical hydrothermal treatment in which the material is exposed to high-pressure saturated steam for a period of time and then suddenly decompressed to achieve the desired effect [6]. It has certain economic advantages and eco-friendly effects on the processing of a large number of raw materials, such as wheat bran, olive leaves, sumac fruits, orange peels, etc. [7]. Application of SE in high-valued utilization of by-products from wheat processing have been proven to have certain benefits, including disrupting porous structures to prompt the release of intracellular active substances, converting structural carbohydrates into soluble dietary fibers, inactivating some enzymes and reducing some anti-nutritional factors, etc. [8]. Some studies have shown that polysaccharides with different molecular weights could be obtained, and soluble dietary fiber could be dissolved more easily compared with the untreated group after SE treatment [9]. Hence, the application of SE in wheat germ modification has been promising, and it is necessary to purify and identify the major polysaccharides of wheat germ modified by SE.

In this paper, the polysaccharide from steam exploded wheat germ (SE-WGP_I_) was purified with DEAE-52 cellulose and Sephadex G-50 gel chromatography. High-performance gel permeation chromatography (HPGPC), infrared spectroscopy (FT-IR), periodate oxidation-Smith degradation, methylation, gas chromatography-mass spectrometry (GC-MS), nuclear magnetic resonance (NMR), Congo red test and thermal analysis were utilized to identify the structural features of purified polysaccharide. The antioxidant activities of purified polysaccharide were further evaluated, including the scavenging performance on DPPH radical and hydroxyl radicals.

## 2. Materials and Methods

### 2.1. Materials and Reagents

Wheat germ was provided by FADA Flour Group in Shandong Province, China. All the chemical reagents and solvents used in this study were analytical grade.

### 2.2. Steam Explosion Process

SE treatment was carried out according to our previous study [10]. Whole wheat germ with a moisture content of 30% (*w*/*w*) was exploded by a laboratory scale SE test bench (Hebi Zhengdao Bioenergy Company, China), in which the pressure was 0.8 MPa and the maintenance time was 5 min. The exploded sample was dried at 60 °C for 24 h and then ground and sieved through a 60 mesh screen. The steam exploded wheat germ (SEWG) was refluxed with petroleum ether for 8 h at 55 °C to decolorize and remove grease and petroleum ether soluble constituents. The samples were dried naturally for subsequent experiments.

### 2.3. Extraction and Purification of Polysaccharide

The extraction procedure of polysaccharide from SEWG was optimized by single factor experiment and response surface experiment with extraction time, temperature, times and ratio of material to liquid as research parameters and polysaccharide yield as detection index. According to the optimum extraction procedure, the defatted SEWG was extracted thrice at 70 °C with distilled water (1: 5, *w*/*v*) under gentle stirring for 30 min each time, and all of the supernatants after centrifugation (4000 rpm/min, 20 min) were concentrated by vacuum rotary evaporator. Then, they were deproteinized 15 times by Sevag reagent [11]. Afterwards, 4 volumes of anhydrous ethanol were added in and stood overnight at 4 °C, and the precipitate was freeze-dried after centrifugation to obtain crude SEWG polysaccharide (SE-WGP). Each SE-WGP sample was weighed with an analytical balance, according to the following equation to calculate the extraction yield (Y):(1)$$Y\%=\frac{\mathrm{weight}\;\mathrm{of}\;\mathrm{SE}-\mathrm{WGP}}{\mathrm{weight}\;\mathrm{of}\;\mathrm{SEWG}}\times 100$$

An amount of 20 mg crude SE-WGP was dissolved in appropriate deionized water to place on a DEAE-Cellulose column (1.6 × 45 cm). The elution was carried out with deionized water and 0.1, 0.3, 0.5, 0.7 mol/L NaCl at a flow rate of 0.8 mL/min [12]. The eluent fractions were measured by phenol-sulfuric acid method, and two fractions were obtained and named as SE-WGP_I_ and SE-WGP_II_. Then, SE-WGP_I_ was further purified with a Sephadex G-50 (1.6 × 45 cm) column and eluted with distilled water at a flow rate of 0.8 mL/min. The main collected parts were dialyzed and lyophilized to obtain purified SE-WGP_I_ for further study.

### 2.4. Chemical Analysis

Phenol sulfuric acid method was used to determine the total sugar content [13]. BCA protein assay method was used to measure the protein content [14], and m-hydroxybiphenyl method was applied to determine the content of uronic acid [15].

### 2.5. Homogeneity and Molecular Weight Determination

The average molecular weight and homogeneity of SE-WGP_I_ were measured by using a Waters HPLC instrument (RID-10A detector) equipped with OHpak SB-804 HQ (8.0 mm × 300 mm) column, taking ultra-pure water as mobile phase. The dissolved sample was filtered by 0.22 μm filter membrane and then injected (20 μL), and analyzed at the elution rate of 0.8 mL/min. The standards were dextrans with different molecular weights (41, 15, 5.0, 2.5, 1.2 and 1 kDa).

### 2.6. Monosaccharide Composition Determination

The determination of monosaccharide composition was conducted based on the published paper with slight adjustments [16]. Briefly, 10 mg SE-WGP_I_ was hydrolyzed in 2 mol/L TFA in a sealed glass tube at 110 °C for 5 h. Then, the sample was reduced with NaBH_4_ after removing excessive TFA with repeated methanol, followed by acidification with acetic acid. The alditol acetate was prepared to be acetylated with pyridine-acetic anhydride (1: 1) at 105 °C for 1 h, and then subjected to Agilent 7890A gas chromatography (GC) system equipped with an OV-1701 capillary column (0.32 mm × 0.5 μm × 30 m). The column temperature was maintained at 150 °C for 1 min, then rose to 200 °C for 10 min with a heating rate 10 °C/min, then increased to 220 °C for 5 min with a heating rate 5 °C/min, and finally increased to 240 °C for 20 min with a heating rate 1.5 °C/min. The temperatures of the injector and the FID detector were set at 250 °C. Arabinose, xylose, mannose, rhamnose, glucose, galactose and fucose were used as monosaccharide standards.

### 2.7. Periodate Oxidation-Smith Degradation

SE-WGP_I_ (20 mg) and 0.03 M NaIO_4_ solution were placed in darkness at 4 °C for reaction. At different time points (0, 4, 8, 12, 24, 48 h), the absorbance at 233 nm of mixture was monitored [17]. When the absorbance ceased descending and remained stable, the reaction was completed and the excess NaIO_4_ was then removed with appropriate amount of ethylene glycol. The formic acid was titrated with 0.099 mol/L NaOH solution to determine its production. The remaining periodate products obtained by the above reactions were dialyzed for 48 h and reduced with 50 mg NaBH_4_ for 24 h. Then, the solution was concentrated and subjected to complete hydrolysis at 105 °C with 2 mol/L TFA for about 5 h. After removing residual TFA and acetylating process, the products were analyzed by GC system.

### 2.8. Methylation Analysis

Methylation analysis of SE-WGP_I_ was performed according to the reported method with some modifications [18]. An amount of 10 mg SE-WGP_I_ was mixed with 50 mg NaOH and dissolved in 2 mL anhydrous DMSO under nitrogen atmosphere and sonicated for 1 h. Methyl iodide (1 mL) was added and performed with sonicating in the dark for 1 h, and then the reaction was stopped by 2 mL distilled water to obtain the methylated polysaccharide. It was extracted four times with dichloromethane, and dried with sodium sulfate and then evaporated to dryness. Afterwards, the methylated SE-WGP_I_ was hydrolyzed and acetylated as mentioned before. The result was analyzed by a GC-MS (VARIAN 4000GC-MS, Palo Alto, CA, USA) system and the chromatographic column was a HP-5 capillary column (30 m × 0.25 mm × 0.25 μm). The column temperature was raised from 100 °C to 280 °C at a heating rate of 10 °C/min, and remained at 280 °C for 12 min.

### 2.9. Infrared Spectrum Analysis

About 1 mg SE-WGP_I_ and 150 mg KBr were mixed and ground quickly to a fine powder. FTIR spectroscopy (IS50, Thermo Fisher, Massa, WA, USA) was measured by the potassium bromide pellet method in a vibrations region of 400–4000 cm^−1^ [19].

### 2.10. NMR Spectrum Analysis

About 20 mg SE-WGP_I_ was dissolved with 0.6 mL D_2_O in an NMR tube and the 1D-NMR (^1^H NMR, ^13^C NMR) and 2D-NMR (^1^H-^1^H COSY, HSQC and HMBC) spectra were obtained using a Bruker Avance Ш (400 MHz) (Cambridge Isotope Laboratories, Inc., Tewksbury, MA, USA).

### 2.11. Congo Red Test

The solution of sample (2.0 mL, 2.5 mg/mL) was mixed with 2 mL of Congo red solution (80 μmol/L). Then, 1.5 mol/L NaOH was continually added to final NaOH concentration of 0, 0.1, 0.2, 0.3, 0.4 and 0.5 mol/L. Full wavelength scanning in the scanning range of 400 to 700 nm was analyzed by a TU-1810 (Ultraviolet spectrophotometer, China), and the maximum absorption wavelengths were recorded [20].

### 2.12. Thermal Analysis

TGA analysis was determined on a TA instrument (model TGA Q50). About 3 mg of SE-WGP_I_ sample was placed in an aluminum crucible and heated from 30 °C to 600 °C at a speed of 10 °C/min [20].

### 2.13. Antioxidant Activities Evaluation

The DPPH radical scavenging activity of SE-WGP_I_ was studied according to previous literature [21] with slight modifications. SE-WGP_I_ was prepared into aqueous solutions with different concentrations (0.2–1.2 mg/mL, 0.5 mL) and mixed with 2.5 mL of DPPH anhydrous ethanol solution (0.4 mM), respectively. The mixtures were shaken thoroughly at 25 °C for 30 min in the darkness, and then were detected at 517 nm by a TU-1810 (Ultraviolet spectrophotometer, China) with ascorbic acid as positive control. The DPPH radical scavenging activity of SE-WGP_I_ was expressed as:(2)$$\mathrm{DPPH}\;\mathrm{radical}\;\mathrm{scavenging}\;\mathrm{activity}\;\left(\%\right)=\left[1-\frac{A_{2}-A_{1}}{A_{0}}\right]\times 100\%$$
where *A*_0_ represents the absorbance of the control substance (water used instead of sample), *A*_1_ represents the absorbance of the sample without DPPH solution under identical conditions and *A*_2_ represents the absorbance of the sample mixed with DPPH solution.

The ability of SE-WGP_I_ to scavenge hydroxyl radicals was measured as by a previous study [22] with slight modification. An amount of 1.0 mL SE-WGP_I_ solutions with different concentrations (0.2–1.2 mg/mL) were blended with salicylic acid-ethanol solution (6 mmol/L, 1 mL) and FeSO_4_ (2 mmol/L, 1.0 mL), respectively. Then, H_2_O_2_ (6 mmol/L, 1.0 mL) was added and the mixed solution was maintained at 37 °C for 30 min. The absorbance of the mixture was immediately measured at 510 nm by a TU-1810 (Ultraviolet spectrophotometer, China) with the positive control of Vc. Hydroxyl radical scavenging activity was calculated in the same way:(3)$$\mathrm{Hydroxyl}\;\mathrm{scavenging}\;\mathrm{activity}\;\left(\%\right)=\left[1-\frac{A_{1}-A_{2}}{A_{0}}\right]\times 100\%$$
where *A*_0_ represents the absorbance of the reagent blank, *A*_1_ represents the absorbance of the sample and *A*_2_ represents the absorbance of the control.

### 2.14. Statistical Analysis

The statistical analysis was performed by one-way ANOVA followed by the Tukey post hoc test using Origin 9.0 statistical software program. Three independent trials were performed, and all the data were expressed as mean ± standard deviation (SD).

## 3. Results and Discussion

### 3.1. Isolation and Purification

The extraction yield of crude polysaccharide prepared from the SEWG was 18.72% (dry weight). In the previous study [5], crude polysaccharides were isolated from defatted wheat germ without steam explosion, and the yield was only 1.16%, indicating that steam explosion technology effectively improved the extraction yield of crude polysaccharides. As we know, the separation mechanism of anion-exchange column chromatography is not only ion exchange, but also adsorption–desorption. Therefore, anion exchange resins are mostly used for the separation of neutral and acidic polysaccharides. Neutral polysaccharides can be obtained by water elution and acidic polysaccharides by salt solution elution [23]. Therefore, SE-WGP was separated and purified by a DEAE-52 column with gradient elution of deionized water, 0.1, 0.3, 0.5 and 0.7 mol/L NaCl. Two independent fractions, SE-WGP_I_ and SE-WGP_II_, were obtained (1.61% losses) with yields of 50.21% and 48.18% (Figure 1). SE-WGP_I_ was eluted by distilled water and further purified (0.98% loss) with a high-resolution Sephadex G-50 size-exclusion column taking distilled water as the eluent. The yield of SE-WGP_I_ was 4.17% from the defatted wheat germ after the dialysis and lyophilization of the first fraction. As shown in Figure 1, SE-WGP_I_ was a homogeneous polysaccharide with a single and tailed peak. A major part of SE-WGP_I_ was collected and detected by the phenol-sulfuric acid method for the determination of 98.8% content. This study mainly focused on the structural characterization of SE-WGP_I_ and its in vitro antioxidant activities. The relevant property of SE-WGP_II_ has been previously reported [5].

### 3.2. Chemical Composition

The content of total sugar in SE-WGP_I_ was determined to be 91.21 ± 3.22% by the phenol-sulfuric acid assay. The content of protein in SE-WGP_I_ was measured to be 1.20 ± 0.09% according to the BCA protein assay method. The content of uronic acid was 1.01 ± 0.02%. Owing to the low content of protein and uronic acid in SE-WGP_I_, it was considered to be a neutral hetero-polysaccharide.

### 3.3. Homogeneity and Average Molecular Weight

The average molecular weight and homogeneity of SE-WGP_I_ was evaluated by HPGPC, as shown in Figure 2a,b. A single and symmetrical narrow peak was detected at about 11.67 min of elution time, and SE-WGP_I_ was possibly homogeneous because of its low polydispersity index (*Mw*/*Mn* = 1.15). The average molecular weight (*Mw*) of SE-WGP_I_ was 5.6 × 10^3^ Da, which was a kind of polysaccharide with lower molecular weight compared to most polysaccharides with hundreds of thousands of Da. For example, the molecular weight of wheat germ polysaccharides reached 4 × 10^3^ kDa when studied by Yun et al. [5].

### 3.4. Monosaccharide Composition

Quantitative monosaccharide analysis of SE-WGP_I_ was performed using the GC method, and the spectrums are shown in Figure 2c (monosaccharide standards) and Figure 2d. The monosaccharides in SE-WGP_I_ were mainly detected to be glucose (Glc), arabinose (Ara) and xylose (Xyl), and the molar ratio was 59.51: 20.71: 19.77. The peak associated with uronic acid was not detected, indicating that the content of uronic acid in SE-WGPI was low. The above results confirmed that SE-WGP_I_ was a homogeneous polysaccharide, in which Glc accounted for the greatest proportion of total monosaccharides. It is easy to find that Ara, Xyl and Glc were found in both wheat germ polysaccharides and bran polysaccharides, but the specific contents were different [5].

### 3.5. Periodate Oxidation-Smith Degradation

According to the periodate oxidation analysis, 1 mol sugar residue consumed 34.5 mmol periodate and produced 0.08 mmol formic acid, which implied that 1-linked or 1,6-linked sugar residue existed in SE-WGP_I_ [24]. However, the consumption of periodate was more than twice the production of formic acid, inferring the possible presence of a large number of 1,2-linked, 1,4-linked or 1,4,6-linked glycosidic bonds. Then, Smith degradation was further performed on SE-WGP_I_, and Xyl, erythritol and glycerol were detected in it (Figure 2e). The existence of Xyl suggested that certain residues of Xyl cannot be oxidized by periodate owing to the existence of 1,3-linked, 1,3,4-linked or 1,3, 6-linked glycosidic bonds. Ara and Glc were not observed, which meant that all the residues of Ara and Glc were oxidized, such as 1-linked or 1,6-linked. Furthermore, the appearance of erythritol suggested the existence of 1,4-linked and 1,4,6-linked glycosidic bonds, and glycerol correlated to the presence of 1-linked, 1,2-linked or 1,6-linked glycosidic bonds in the backbone of SE-WGP_I_ [25].

### 3.6. Methylation Analysis

As shown in Table 1, the methylation analysis demonstrated that SE-WGP_I_ had five partially O-methylated alditol acetate (PMAAs) peaks. By comparing with standard data and other papers [17], it could be discovered that SE-WGP_I_ contained 1,4,5-tri-O-acetyl-2,3-di-O-methyl-Arabitol (A, 18.56 mol%), 1,3,5-tri-O-acetyl-2,4-di-O-methyl-xylitol (E, 19.28 mol%), 1,5-di-O-acetyl-2,3,4,6-tera-O-methyl-D- glucitol (C, 20.05 mol%), 1,5,6-tri-O-acetyl-2,3,4-tri-O-methyl-D-glucitol (D, 20.34 mol%) and 1,4,5,6-tera-O- acetyl-2,3-di-O-methyl-D-glucitol (B, 21.77 mol%), which were identified as 1,5-linked L-Araf, 1,3-linked D-Xylp, T-D-Glcp, 1,6-linked D-Glcp and 1,4,6-linked D-Glcp, respectively. In all PMAA mass spectra, there was a base peak of *m*/*z* 43 formed by the loss of acetyl ions (CH_3_CO^+^). The C2 and C3 in 1,5-linked L-Araf were all methylated carbon. The methylated C-C was easy to break, mainly producing *m*/*z* 117, *m*/*z* 189, *m*/*z* 161 and *m*/*z* 145 ion fragments. The ion fragments of *m*/*z* 161 were degraded into m/z 101 secondary ion fragments, which was eventually degraded into *m*/*z* 71, *m*/*z* 43 and other smaller ion fragments. In addition, *m*/*z* 189 was further broken into *m*/*z* 129. In 1,4,6-linked D-Glcp, C2 and C3 were methylated carbon, C4 was acetylated carbon and C2-C3 was easier to break than C3-C4. When the break occurred in C2-C3, ion fragments of *m*/*z* 117 and *m*/*z* 261 were produced, and *m*/*z* 261 was further broken into secondary ion fragments of *m*/*z* 201. When the fracture occurred in C3-C4, the ion fragment of *m*/*z* 161 was produced and further split into the secondary ion fragment of *m*/*z* 101. In T-Glcp, *m*/*z* 117 and *m*/*z* 205 were ion fragments produced by the fracture between C2-C3, and *m*/*z* 205 split into ion fragments *m*/*z* 145. When the fracture occurred between C3-C4, the ion fragments of *m*/*z* 161 and *m*/*z* 162 were produced, which were broken into the secondary ion fragments of *m*/*z* 101 and *m*/*z* 129. These secondary ion fragments were further broken into ion fragments of *m*/*z* 71 and *m*/*z* 43. In 1,6-linked D-Glcp, C2, C3 and C4 were all methylated carbon, so the fracture occurred between C2-C3 and C3-C4. When the fracture occurred in C2-C3, the main ion fragments were *m*/*z* 117 and *m*/*z* 233, and *m*/*z* 233 was further broken into *m*/*z* 145. When the fracture occurred in C3-C4, the main ion fragments were *m*/*z* 161 and *m*/*z* 189, which were broken into *m*/*z* 101 and *m*/*z* 129, respectively. In 1,3-linked D-Xylp, C2 and C4 were methylated carbon. When the break occurred in C1-C2, ion fragments of *m*/*z* 233 were produced. When the break occurred in C2-C3, *m*/*z* 118 was produced. When the fracture occurred in C3-C4 and C4-C5, the ion fragment of *m*/*z* 117 and *m*/*z* 234 was produced and further split into the secondary ion fragment. This agrees with the results of periodate oxidation-Smith degradation analysis. It can be seen from Table 1 that glycosidic bonds related to Glc were 62.16% of all monosaccharides, being consistent with the analysis in Section 3.4. The above results revealed that SE-WGP_I_ contained the linear polysaccharides with branching chain, and the main backbone of SE-WGP_I_ contained 2,3-Me2-Araf, 2,4-Me2-Xylp, 2,3-Me2-Glcp, 2,3,4,6-Me4-Glcp and 2,3,4-Me3-Glcp.

### 3.7. FT-IR Spectrum

As shown in Figure 3, the strong and broad band at around 3412 cm^−1^ was the stretching vibration of OH due to intra- and inter-molecular hydrogen bands. The absorption at 2932 cm^−1^ was due to the C-H stretching vibrations of CH_2_ groups of free sugars [26]. The strong band at 1653 cm^−1^ indicated the C=O bonds of a carbonyl group. These bands at 3412, 2932 and 1653 cm^−1^ were characteristics of a carbohydrate ring, and C-H led to a weak absorption at approximately 1412 cm^−1^ because of the deformation vibration. The weak band at 1248 cm^−1^ was assigned to the absorption of CH_3_ from the acetyl group. Three peaks at 1150, 1045 and 1000 cm^−1^ implied the stretching vibrations of a pyranose ring [27]. The weak absorption at 888 cm^−1^ and 838 cm^−1^ represented the presence of *β*-pyranoside and α-pyranoside, respectively [28]. As a result, the spectra indicated that the major groups of SE-WGP_I_ were not affected by the purification process, and SE-WGP_I_ contained α-type and *β*-type glycosidic linkages with a pyranose ring.

### 3.8. NMR Spectroscopy

The structural characteristics of SE-WGP_I_ were further elucidated by 1D- (^1^H, ^13^C NMR) and 2D-NMR (COSY, HSQC, HMBC spectra). The ^1^H NMR spectrum (Figure 4a) showed that the chemical shifts of anomeric proton in this study ranged from δ 4.90 ppm to 5.32 ppm, including weak signals at δ 5.02 and 5.16 ppm and strong signals at δ 4.90, 5.28 and 5.32 ppm. The presence of these five signals proved that SE-WGP_I_ contained both *α*-configuration and *β*-configuration, and it was confirmed by the conclusion of FT-IR [29]. In Figure 4b, the distribution of the anomeric carbon ranged from δ 91.90 to 107.44 ppm in the ^13^C NMR spectrum. The signal at δ 91.90, 98.06, 98.53, 99.96 and 107.44 ppm indicated that five types of linkage patterns existed in SE-WGP_I_, and this was in good accordance with the conclusions of methylation.

In the HSQC spectrum of SE-WGP_I_ (Figure 4d), δ 5.02/107.44, 5.28/99.96, 5.16/91.90, 5.32/98.06 and 4.90/98.53 ppm were detected in the anomeric region and sequentially marked as sugar residues A, B, C, D and E. According to the results of the NMR analysis, methylation analysis and literature data, it could be deduced that A, B, C, D and E were identified as →5)-*α*-L-Ara*f* (1→, →4,6)-*α*-D-Glc*p* (1→, *α*-D-Glc*p* (1→, →6)-*α*-D-Glc*p* (1→ and →3)- *β*-D-Xyl*p* (1→, respectively. The chemical shifts of H and C for all the residues are summarized in Table 2.

Figure 4e shows the spectrum of HMBC from which the glycosidic linkages between sugar residues can be deduced in more detail. The cross-peaks at δ 3.62/99.96 (CH6/BC1), 5.32/72.73 (DH1/BC2) and 4.90/70.40 (EH1/DC4) indicated the presence of →4,6)-*α*-D-Glc*p-*(1→6)-*α*-D-Glc*p*(1→3)-*β*-D-Xyl*p*(1→ in SE-WGP_I_. Based on the methylation analysis, the cross-peak δ 3.62/99.96 (CH6/BC1) and 3.62/71.06 (BH6/CC4) should also be attributed to the correlation between residue B and C, suggesting the residue B was terminated with residue C, showing that there was a sequence of *α*-D-Glc*p* (1→4,6)-*α*-D-Glc*p* (1→ in SE-WGP_I_. Therefore, the backbone of SE-WGP_I_ was mainly made up of →4,6)-*α*-D-Glc*p*(1→6)-*α*-D-Glc*p* (1→3)*β*-D-Xyl*p* (1→5)-*α*-L-Ara*f* (1→ and the side chain was *α*-D-Glc*p* (1→ linked at the C4-position of →4,6)-*α*-D-Glc*p* (1→. A possible predicted repetitive unit structure of SE-WGP_I_ is proposed in Figure 4f.

### 3.9. Conformational Structure

The Congo red assay has been increasingly used due to its simplicity of operation and instrument independence, for example, to identify the triple helix conformation of polysaccharides [30]. As an acid dye, Congo red formed complexes with polysaccharides containing the triple helix configuration. In a certain concentration range, the maximum absorption wavelength (λ_max_) of the complexes showed a bathochromic shift, thus forming a metastable region in comparison with Congo red (control group) [31]. Figure 5a demonstrates the changes in λ_max_ of Congo red and polysaccharide complex with 0~0.5 mol/L NaOH concentrations. It can be seen from the figure that the λ_max_ of the sample had a red shift phenomenon at all alkaline concentrations. Therefore, we speculated that SE-WGP_I_ might have a triple helix conformation in solution. At the same time, the view that polysaccharides with three-helix structures usually had higher biological activities has been confirmed by many studies [32].

### 3.10. Thermal Analysis

It is very important to analyze the thermodynamic properties of polysaccharides. TGA results showed the thermal degradation characteristics and weight loss of degraded products during the temperature changing process. In Figure 5b, DTG curves showed a slight mass loss ranging from 30 °C to 200 °C, resulting from the loss of free and bound water in SE-WGP_I_. A large weight loss from around 200 °C to 600 °C was observed due to the degradation of the polysaccharide structure by dehydroxylation or deoxylation [33].

The measured chemical structure of the polysaccharide determined its thermodynamic properties, both of which were influenced by the modification method. According to our previous study, the degradation temperature of the polysaccharide extracted from wheat germ was 290.03 °C, and the weight loss was 88.42% at 600 °C [5]. In this study, the degradation temperature of SE-WGP_I_ was determined to be 234.05 °C, and the weight loss was 70.25% at 600 °C. This indicated that the thermodynamic properties of polysaccharides were obviously different before and after steam explosion treatment. The low degradation temperature of SE-WGP_I_ could be attributed to the hydrothermal degradation of polysaccharides caused by steam explosion, resulting in the generation of some substances that were easy to be degraded. The low weight loss of SE-WGP_I_ could be ascribed to the polymerization or aggregation of some macromolecular substances, which had strong heat resistance and were not easy to decompose at 600 °C. The results indicated that SE-WGP_I_ possessed superior thermal stability. From the perspective of food industry production, good thermal stability is conducive to maintaining the nutritional quality of materials during thermal processing such as sterilization and baking [34].

### 3.11. Antioxidant Activities

In many studies on structure characterization and functional activities of polysaccharides from wheat bran, researchers have proposed and verified that wheat bran polysaccharides had good in vivo and in vitro antioxidant activities [35,36]. Inspired by these articles, we tentatively evaluated the in vitro antioxidant activities of wheat germ polysaccharides.

DPPH radical scavenging activity of SE-WGP_I_ with different concentrations was investigated as depicted in Figure 5c. SE-WGP_I_ showed a dose-dependent radical scavenging capacity at 0.2~1.2 mg/mL. It presented the highest DPPH radical scavenging activity of 67.41 ± 1.16% at 1.2 mg/mL. In a previous report [37], two polysaccharides extracted from wheat bran showed weak DPPH radical scavenging activity with EC_50_ estimated at 9.6 and 4.2 mg/mL, respectively. In contrast, SE-WGP_I_ had an obvious scavenging result on DPPH radicals with the EC_50_ of 0.81 mg/mL. It was also found in the literature that the DPPH radical scavenging ability of SE-WGP_I_ was higher than that of the wheat germ polysaccharide with an average molecular weight of 2.99 × 10^5^ Da, and its scavenging rate at 4 mg/mL was 50.21% [35]. SE-WGP_I_ contained a low amount of protein, which has been reported to play a significant effect on the overall radical scavenging activity of polysaccharides [38].

In Figure 5d, when the concentration of SE-WGP_I_ was the highest, the scavenging rate of this polysaccharide on hydroxyl radicals reached 40.21%, which was also dose-dependent. Meanwhile, the hydroxyl radical scavenging ability of SE-WGP_I_ was similar to that of the wheat germ polysaccharide, with an average molecular weight of 2.99 × 10^5^ Da [35]. The most likely pathway of antioxidant activity of the polysaccharide on hydroxyl radicals was that the polysaccharide could provide hydrogen to stabilize free radicals and/or directly react with them to terminate the free radical chain reaction [36]. The above results indicated that SE-WGP_I_ in this study had relatively good antioxidant activity.

## 4. Conclusions

This is the first report on the purification and structural characterization of the hetero-polysaccharide from wheat germ modified by SE treatment. SE-WGP_I_, as a novel water-soluble polysaccharide, was composed of arabinose, xylose and glucose with a molar ratio of 20.71: 19.77: 59.51. Based on the results of FT-IR, periodate oxidation-Smith degradation, methylation and NMR spectroscopy, the backbone of SE-WGP_I_ was →4,6)-*α*-D-Glc*p* (1→6)-*α*-D-Glc*p* (1→3)-*β*-D-Xyl*p* (1→5)-*α*-L-Ara*f* (1→ and the side chain was *α*-D-Glc*p* (1→ linked at the C4-position of →4,6)-*α*-D-Glc*p* (1→. The result of the Congo red test suggested that SE-WGP_I_ possibly has a triple helix conformation, and it had relatively good thermal stability according to the TGA analysis. In addition, SE-WGP_I_ showed valid in vitro antioxidant activities in a dose-dependent manner. This study provides a theoretical basis for further research into the structure–activity relationship of SE-WGP_I_. The bioactivity of SE-WGP_I_ is currently being studied in our laboratory.

## Figures and Tables

**Figure 1 foods-11-01245-f001:**
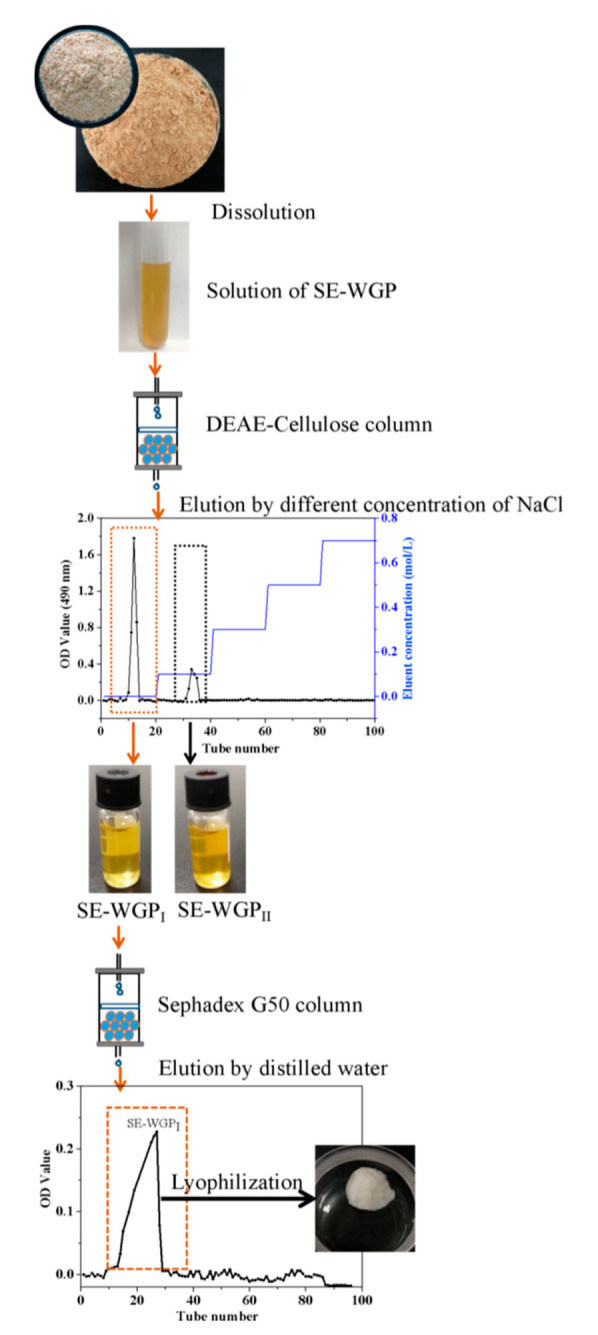
The separation flow diagram of SE-WGP_I_.

**Figure 2 foods-11-01245-f002:**
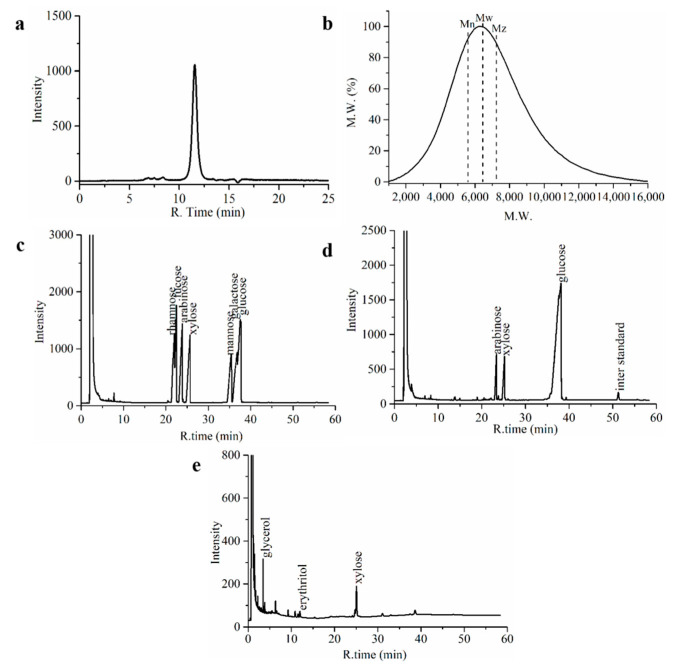
Molecular weight distribution of SE-WGP_I_ (**a**). Differential mass distribution curve of SE-WGP_I_ (**b**). GC chromatography of monosaccharide standards (**c**). SE-WGP_I_ (**d**) and Smith degradation of SE-WGP_I_ (**e**).

**Figure 3 foods-11-01245-f003:**
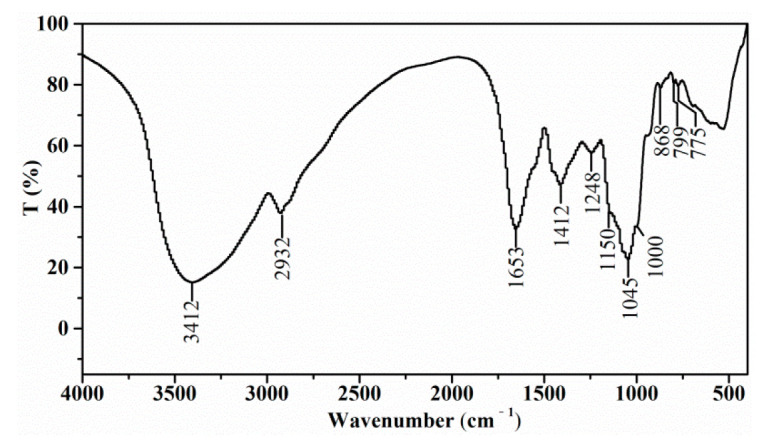
The Fourier transforms infrared spectrogram of SE-WGP_I_.

**Figure 4 foods-11-01245-f004:**
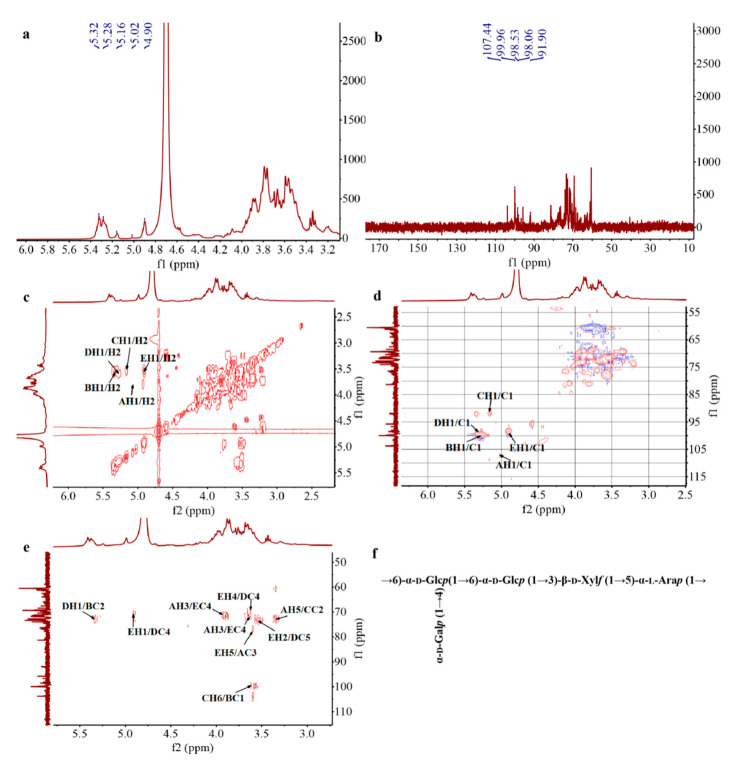
^1^H NMR (**a**), ^13^C NMR (**b**), COSY (**c**), HSQC (**d**), HMBC (**e**) spectra and the proposed structure of SE-WGP_I_ (**f**).

**Figure 5 foods-11-01245-f005:**
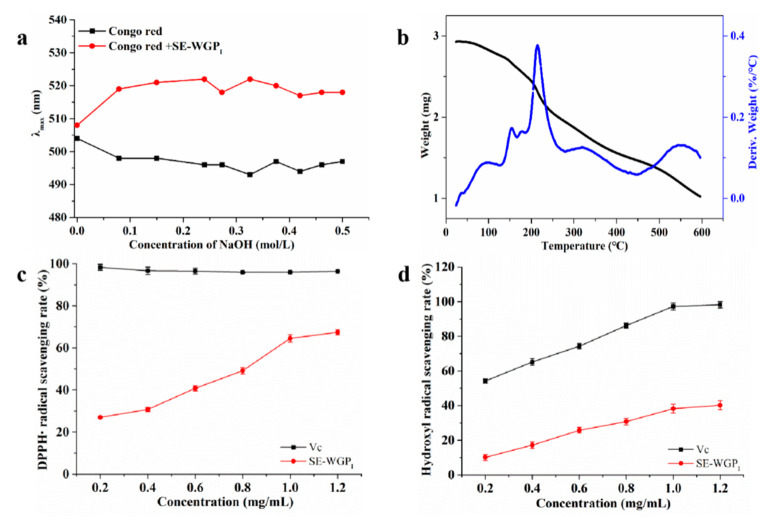
Maximum absorption (λmax) of Congo red and Congo red + polysaccharide at different concentrations of NaOH (**a**). The TGA curves of SE-WGP_I_ of different moisture contents (**b**). DPPH radical scavenging activity of SE-WGP_I_ (**c**). Hydroxyl radical scavenging activity of SE-WGP_I_ (**d**).

**Table 1 foods-11-01245-t001:** Methylation analysis of SE-WGP_I_.

Residues	Retention Time (min)	PMAAs	Type of Linkage	Molar Ratio (mol %)	Major Mass Fragments (*m*/*z*)
A	29.25	1,4,5-tri-O-acetyl-2,3-di-O-methyl-Arabitol	1,5-linked L-Ara*f*	18.56	43, 59, 71, 87, 101, 117, 129, 173, 189
E	29.60	1,3,5-tri-O-acetyl-2,4-di-O-methyl-xylitol	1,3-linked D-Xyl*p*	19.28	43, 59, 71, 87, 101, 117, 129, 189, 201, 233
C	30.87	1,5-di-O-acetyl-2,3,4,6-tera-O-methyl-D-glucitol	T-D-Glc*p*	20.05	43, 59, 71, 87, 101, 113, 117, 129, 145, 161, 162, 205
D	32.54	1,5,6-tri-O-acetyl-2,3,4-tri-O-methyl-D-glucitol	1,6-linked D-Glc*p*	20.34	43, 59, 71, 87, 101, 117, 129, 145, 162, 173, 189,233
B	33.24	1,4,5,6-tera-O-acetyl-2,3-di-O-methyl-D-glucitol	1,4,6-linked D-Glc*p*	21.77	43, 59, 71, 87, 101, 117, 129, 161, 173, 201, 217, 233, 261

**Table 2 foods-11-01245-t002:** Chemical shifts of resonances in the ^1^H and ^13^C NMR spectra of SE-WGP_I_.

Sugar Residue		Chemical Shift (ppm)					
		H1/C1	H2/C2	H3/C3	H4/C4	H5/C5	H6/C6
A	→5)-α-L-Ara*f* (1→	5.02/107.44	3.84/81.22	3.90/77.17	3.73/84.07	3.38/67.24	-
B	→4,6)-α-D-Glc*p* (1→	5.28/99.96	3.64/72.73	3.23/73.71	3.61/78.04	3.79/70.98	3.62/66.90
C	α-D-Glc*p* (1→	5.16/91.90	3.54/72.99	3.97/70.28	3.79/71.06	3.80/70.98	3.62/69.90
D	→6)-α-D-Glc*p* (1→	5.32/98.06	3.63/72.30	3.98/73.09	3.65/70.40	3.37/72.30	3.61/66.10
E	→3)-β-D-Xyl*p* (1→	4.90/98.53	3.54/72.50	3.80/81.07	3.62/70.70	3.61/66.01	-

## Data Availability

Data is contained within the article.

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
