# Peer review of "Isolation and Purification, Structural Characterization and Antioxidant Activities of a Novel Hetero-Polysaccharide from Steam Exploded Wheat Germ"

_foods, 2022, doi:10.3390/foods11091245_

Round 1

Reviewer 1 Report

The paper looks very interesting by means of full basic characterisation of polysaccharides extracted from wheat germ (steam exploded). According to fully understand the structure of investigated polymer all analyses needed has been done. As a result a full description of polymer has been obtained. There is no doubts about experiment design and methodology.

My doubts and suggestion that needs to be clarified are as follows:

  1. I understand the structure of polymer extracted. But there is lack of information (even basic) the structure of the polymers in germ before steam explosion has been performed. Without comparison we do know nothing on the influence of hydrothermal treatment on polymer structure.
  2. Extraction procedure is easy and clear. Did authors performed any optimisation of the procedure by means of time, pH or other factors?
  3. How the yield was calculated? Based on polysaccharide content in raw material? How it was established? It is a difficult problem since nothing is known on polysaccharides in wheat germ before and after steam explosion
  4. According to obtained data (NMR, GC) the polymer is an example of the branch macromolecule. So I wonder why dextran standards were in use. In this cases maybe pullulan standards are more useful.
  5. I suggest to add differential mass distribution instead of SEC (GPC) chromatograms. It will be easier to follow the discussion.

According to all this comment I designate the paper as needed minor revision.

Reviewer 2 Report

The manuscript by Hu et al. entitled "Isolation and purification, structural characterization and anti-oxidant activities of a novel hetero-polysaccharide from steam exploded wheat germ" details an attempt to elusidate the structure and some characteristics of an isolate (SE-WGPI) from steam exploded wheat germ. The manuscript is fairly well written and is easy to follow. It however need some editing to improve the english language. 

The results are fairly well presented, although the description of the experiments needs improvement. Furthermore there is a lack of references to specific methods and research, including the authors own research (top page 2).

There are several questionable choices for experiments and interpretations of data that needs to be adressed. 

1) The authors use ion-exchange chromatography to purify the specific fraction in focus. As the purified polymer is neutral under the conditions used this seems a bit strange and the polymer should elute with the void volume of the column. Is this the case? If so it elutes with all other neutral compounds in the eluate and most likely is not pure. 

2) The polydispersity index of SE-WGPI is very low (1.05) and highly unlikely for a carbohydrate polymer. In biological systems only peptides, proteins, DNA and RNA presents such low DPI. The evidence for the purity of the polymer is thus highly dubious. 

3) A single study using congo red complexation with the SE-WGPI leads to the conclusion that the polysaccharide adopts a triple-helix conformation. This is by all means substantiated and must be considered as pure speculation.

4) The morphological studies of SE-WGPI using SEM and AFM bring no useful information to the elusidation of the structure of the polymer. It only shows the morphology of a dries solution of SE-WGPI (e.g. how the fraction dries up into patches/string formed structures) and does not reveal anything about the structure of the polymer. The resolution of the methods does not allow any scrutiny of the molecular structure of the polymers in the sample. 

5) Thermal analysis does not bring forward important information about SE-WGPI and the data does not fully make sense. The DSC study of the effect of moisture content is also hard to follow. Here the change in DH is claimed to be related to a gel network, although the moisture content needed for the formation of a gel network is by far not reached. 

6) Why was antioxidant activity assessed? Is it surprising that the polymer probably has antioxidant like activities common comparable to other carbohydrates? As the antioxidant activities of SE-WGPI are not compared to other carbohydrates and only ascorbic acid, there is not reason to speculate that the results are surprising or even relevant.

In conclusion, there are many unclear and unsubstantiated claims in the manuscript. Although the authors attempts to link the study to Food, the links seems a bit far fetched (some dubious results on thermal stability and antioxidant effects). A reconsidered and rewritten manuscript may be more suitable for a more specialized journal such ad Carbohydrate Polymers or Carbohydrate Research.        

3)  

Round 2

Reviewer 2 Report

The revised version of "Isolation and purification, structural characterization and antioxidant activities of a novel hetero-polysaccharide from steam exploded wheat germ" by HU et al. is significantly improved. Overall the authors have answered satisfactorily to the questions raised, but may still need to validate their results in terms of what is indistibutable facts and what is suggestions/speculation. E.g. in the case of triple-helix detection using congo-red, the authors should change the sentence "Therefore, it was concluded that SE-WGPI had a triple helix conformation in solution" to  "Therefore, we speculated that SE WGP I might have a triple helix conformation in solution". 
